# A Data Driven Approach for Raw Material Terminology

Olivera Kitanović [1,*,†], Ranka Stanković [1,†], Aleksandra Tomašević [1,†], Mihailo Škorić [1,†], Ivan Babić [2,†] and Ljiljana Kolonja [1,†]

1   Faculty of Mining and Geology, University of Belgrade, 11000 Belgrade, Serbia;
    ranka.stankovic@rgf.bg.ac.rs (R.S.); aleksandra.tomasevic@rgf.bg.ac.rs (A.T.);
    mihailo.skoric@rgf.bg.ac.rs (M.Š.); ljiljana.kolonja@rgf.bg.ac.rs (L.K.)
2   Department for Informatics and Computing, University of Criminal Investigation and Police Studies,
    11000 Belgrade, Serbia; ivan.babic@mup.gov.rs
*   Correspondence: olivera.kitanovic@rgf.bg.ac.rs; Tel.: +381-11-3219-212
†   These authors contributed equally to this work.

**Abstract:** The research presented in this paper aims at creating a bilingual (sr-en), easily searchable, hypertext, born-digital, corpus-based terminological database of raw material terminology for dictionary production. The approach is based on linking dictionaries related to the raw material domain, both digitally born and printed, into a lexicon structure, aligning terminology from different dictionaries as much as possible. This paper presents the main features of this approach, data used for compilation of the terminological database, the procedure by which it has been generated and a mobile application for its use. Available (terminological) resources will be presented—paper dictionaries and digital resources related to the raw material domain, as well as general lexica morphological dictionaries. Resource preparation started with dictionary (retro)digitisation and corpora enlargement, followed by adding new Serbian terms to general lexica dictionaries, as well as adding bilingual terms. Dictionary development is relying on corpus analysis, details of which are also presented. Usage examples, collocations and concordances play an important role in raw material terminology, and have also been included in this research. Some important related issues discussed are collocation extraction methods, the use of domain labels, lexical and semantic relations, definitions and subentries.

**Keywords:** raw material; mining; terminology; dictionary; terminology application; mobile application; digitization; lexical data; corpus data; linguistic linked open data

## 1. Introduction

During the last decade, lexicography entered a new era due both to rapid development of advanced computational methods and availability of previously unseen abundance of language data in different modalities. These developments have opened new opportunities for producing modern Serbian monolingual and bilingual dictionaries, which will overcome the shortcoming of existing ones, characterized by obsolescence of macrostructure, microstructure and data presentation, frequent inaccuracy of translation, visual and typographic monotony, and a neglect of needs of potential users [1]. These new, modern dictionaries will enable potential users, including students, translators, teachers, researchers and other interested parties, to find all information on formal and contextual properties of words and their interrelationships, in one place. In addition to new human readable monolingual and bilingual dictionaries, machine readable dictionaries of both kinds are also needed. In this situation, a comprehensive approach, combining all available resources, which can be used for producing various types of dictionaries, especially in specialized and terminological domains, seem to be the optimal solution.

According to the findings of the Elexis project [2], the main positive changes in lexicography in the last 10–15 years are mostly related to digitisation and automation of

lexicographic work, online publishing (moving from paper to online) and, with the beginning of the corpus era, by access to corpora supported by (semi)automatic extraction of terms. Automatic data extraction comprises data that is automatically obtained from corpora of authentic language use, which is then subjected to lexicographers' post-processing or included, as is, in the published dictionary, but marked as automatically derived from corpus data. It should be noted that data derived from existing lexical databases and dictionaries should be considered as reuse of data. One of the issues related is the processing and representation of terminological phrases, or multiword expressions (MWEs), ranging from compound nouns (e.g., nickname) to complex phrasal verbs (e.g., give up) and idiomatic expressions (e.g., break the ice), which has remained a challenge over the past 20+ years [3]. In our research we focused on semantically transparent terminological phrases, as well as terminological phrases that result in a meaning shift. Some frequent syntactic patterns and translation options will be discussed. In our approach we will use a combination of: reuse of data, automatic extraction and manual postediting.

The advantage of using online platforms, which offer the possibility of regular updates and a more effective collaboration via the internet, as well as the use of mobile devices were highlighted in literature [4]. The impact of mobile devices as a distribution method is immense, and a mobile-first approach is now instrumental. The general shift towards (mobile) life online brought a clear realization that "printed lexicography"—in general terms—is a thing of the past, and this also turned its business side upside down [5].

Wide adoption of mobile devices has created new ways of learning through interaction and communication and they are becoming integrated in the lives of today's students, enhancing mobility of the learning process. Thus, for example, Language for Specific Purposes (LSP) dictionaries are now being produced at the university level using mobile LSP lexicography. One such dictionary called MobiLex was produced at the Stellenbosch University in South Africa to enhance teaching and learning of historical terms, with favorable pedagogical consequences regarding the learning of such terms. Trends and developments in technology offer the possibility of changing the face of lexicographical support in a mobile environment, from a pedagogical perspective [6].

Big data analysis methods have opened new possibilities for analyzing corpora, which contain large amounts of textual data. Thus, for example, Chen et al. [7] propose a novel statistic-based corpus machine processing approach to refine big textual data, to be used for ESP (English for Specific Purposes). The approach is based on establishing a function word list and embedding it into the program, in order to refine the word list and keyword list. The aim is to enhance the efficiency of corpora processing, starting from preparatory work, followed by generating raw data, optimizing the process, and ending by generating refined data. COVID-19 news reports are used as a simulation example of big textual data and applied to verify the efficacy of the machine optimizing process.

Electronic lexicography offers important possibilities in comparison to the traditional approach. Examples of usage may be extracted from original texts and linked to dictionary entries. There are practically no limitations to the amount of data that can be added, including multimedial data, which results in better quality data. Various search options and different possibilities of database organization contribute to the efficiency of access. Dictionaries can be easily customized for specific needs of users' groups. Electronic lexicography also enables hybridization, by breaking limits between different types of language resources—for example, dictionaries, encyclopedias, term banks, lexical databases, translation tools and the like. Finally, active user involvement is possible, by enabling collaborative or community-based input to dictionaries [8].

This paper presents a data driven approach aimed at using opportunities offered by electronic lexicography, as well as various available techniques of Natural Language Processing (NLP), to develop a semi-automatic pipeline for dictionary production. The approach is focused on raw material terminology, with an emphasis on terminology related to the mining industry, as a case study, the main goal being to cover Serbian and bilingual English-Serbian terminology in the raw material domain, within a system that can be

used for developing web and mobile dictionary applications. In developing this system, a data driven approach is adopted, relying on available textual, lexical and terminological resources, both in printed and electronic form. Within the development of this system, printed resources, the paper dictionaries covering raw material terminology, were subjected to systematic extensive digitisation.

In this approach, besides compiling a comprehensive multilingual lexical database of raw material terminology, lexicographic methods for automatic knowledge extraction are used, including corpus data analysis, automatic data extraction, editing and publishing extracted data in (online) dictionaries. Using extracted lexicographically relevant data (lemma lists, example sentences, collocations) as complementary resources in electronic dictionaries is known as the one-click dictionary or push-pull dictionary model, which is used, for example, in the Sketch-engine [9] for several languages, but has not yet been used for Serbian.

A similar approach to the one outlined in this paper was applied in development of the Sõnaveeb language portal of the Institute of the Estonian Language, which contains data from a number of dictionaries and termbases, with a total of 200,000 Estonian headwords with collocations, etymology, multi-word expressions, etc. The main issues to be resolved in their approach were the consistency of information, deduplication, parsing data fields containing more than one data element, moving from annotating form (e.g., italics) to annotating content (e.g., a citation) [10].

López-Úbeda et al. [11] present another interesting approach, which also combines different NLP techniques to develop a system for identification of biomedical terms in textual documents written in Spanish. The approach was applied for recognizing biomedical entities in various types of texts, including different knowledge resources (MedLine Encyclopedia, International Classification of Diseases, Unified Medical Language System, etc.). Although the tool developed within their approach has been developed for Spanish, the authors plan to expand its usability by incorporating multilingual support in the future, thus enabling it to be extrapolated to other languages.

The web and mobile applications for raw material terminology developed as a result of our approach are primarily intended for students and engineers involved in the raw material industry, as an aid in mastering terminology. They offer both English-Serbian and Serbian-English terminology, developed, inter alia, on using a comprising a variety of literature from the field of raw materials. Existing terminological dictionaries and general language dictionaries served as control dictionaries (listed in the bibliography and described in Sections 2.1 and 3.1). The developed dictionaries are not comprehensive, but rather contain basic terminology from various raw material subdomains (areas), needed to make reading professional literature easier, academic writing purposes and to improve communication among professionals in the raw material industry. In addition to core raw material terms, some technical and academic vocabulary is also introduced, that is, words that often appear in professional literature.

The developed dictionaries are not prescriptive, as they do not prescribe how the terminology "should" be systematized, but rather record the terms in use. Therefore, they feature synonyms and also record technical jargon and localisms next to standard terminology. For example, *'rotorni bager'*, namely, *'bucket wheel excavator'*, is recorded on the Serbian side together with *'glodar'*, a jargon term, literally translated as *'gnawer'*. The publication of the dictionaries as a mobile app is especially important in view of the fact that the job of an engineer dealing with raw materials usually involves frequent field work and staying in the field for prolonged periods.

Section 2 gives an overview of available resources: paper and electronic dictionaries, as well as corpora used. Section 3 outlines preparation of resources, which includes digitization of paper dictionaries, enlargement of corpora, adding domain terms to general purpose morphological e-dictionaries and extraction of bilingual lists. The process of terminology compilation, from the perspective of monolingual and bilingual extraction, a well as the web and mobile form of the dictionary are given in Section 4. The last section

offers a discussion, concluding remarks and outline of future plans for improvements and application in other areas.

## 2. Available (Terminological) Resources

Our approach relies heavily on available resources, both in paper and electronic form, such as traditional, paper dictionaries used in raw material industry, termbases covering raw material terminology, corpora of texts from the raw material domain as well as general-purpose electronic dictionaries of Serbian. This section offers an overview of these resources.

### 2.1. Paper Dictionaries for Raw Material Domain

The Bureau of Mines (U.S. Department of the Interior) had pioneered efforts in mining terminology, beginning in 1918 with Fay's "Glossary of the Mining and Minerals Industry", and continuing by the 1968 publication of "A Dictionary of Mining, Minerals, and Related Terms" (DMMRT). In this 5-year project, more than 100 bureau personnel (engineers, scientists, and editors) were involved in the technical review and publication production process of the dictionary, with 28,750 terms explained by 37,180 sense definitions [12]. This dictionary has been used for several decades at the University of Belgrade Faculty of Mining and Geology (UBFMG), and it is the main dictionary covering mining terminology in English in our approach. Online version of dictionary is published on The Edumine platform that provides professional development training for people in the mining industry [13].

A multilingual "Mining dictionary: Serbo-Croatian: English: French: German: Russian" (MD), containing 16,500 terms related to underground and surface excavation, preparation of mineral raw materials, as well as rock and soil mechanics in five languages was published in 1970 [14]. This dictionary also contains terms from the fields of geology, metallurgy, electrical engineering, mathematics with computational methods, and civil engineering, to the extent they are related to mining. Each term entry has a Serbian headword, sometimes followed by synonyms, which is aligned with translations in four languages—English, French, German, and Russian. The interconnection of all five languages is given by additional indexes. Term entries do not have definitions nor usage examples. the dictionary being almost 50 years old, many terms are outdated, while some new terms are missing. This dictionary was our main source for extracting terminological equivalents in Serbian and English.

The first terminological "English-Croatian-Serbian Petroleum Dictionary" for the field of petroleum engineering [15] was followed, after 30 years, by the "English-Croatian encyclopedic dictionary of oil and gas exploration and production" [16], which is used both in Croatia and Serbia. With 12,200 definitions and 7100 terms, it contains a comprehensive vocabulary of both scientific and professional terms used by scientists, experts and students in the area of exploration and production of oil and gas, but also petroleum geology, geophysics, development deposits, drilling and equipping wells, ecology and other disciplines.

There is also a small bilingual dictionary of mineral processing [17] with 2415 translation pairs, in both directions, English to Serbian and Serbian to English, but also without definitions. Finally, a glossary of mineral processing terms with 1400 definitions in Serbian is used at the UBFMG, although it was not officially published [18].

All these dictionaries, and a number of other dictionaries, a total of 22, have been digitized for the purpose of our approach.

### 2.2. Digital Resources in Raw Material Domain

The development of digital resources for raw material terminology has been an ongoing activity at the UBFMG for several years now. It started with research related to the development of an ontology of mining equipment [19], in line with other research aimed at development of bilingual lexical resources [20]. The focus was then turned to development of termbases for the general field of mining engineering, and their transformation from their

initial custom in-house scheme into the TermBase eXchange (TBX) Standard [21]. Another terminological resource, mostly handcrafted, was also developed to support knowledge management in specific subfields of mining engineering, such as mining equipment, mine safety and geostatistics [22]. A thesaurus of mining terminology is available online, but it is not systematically updated. Moreover the application has no new features, and it is not responsive. A modest experiment was made with developing students' vocabulary related to raw materials through flashcards and L1 in the CLIL Classroom [23], but it was not finalized with publicly available online resources.

Three digital resources already developed at UBFMG were included in our approach, two termbases, Termi [24], and GeoliSSTerm [25], and one ontology, Rudonto [26]. Termi supports development of terminological dictionaries in various fields (mathematics, computer science, raw material, library science, computational linguistics, power engineering, etc.) [27,28], and it has been selected as the most suitable resource to be used for the comprehensive multilingual lexical database of raw material terminology, while the remaining two resources have been incorporated in the dictionary production pipeline.

For systematic development of raw material terminology, textual resources, namely, bilingual libraries and corpora are also needed. Thus, articles from the scientific journal Underground Mining, published both in Serbian and English, stored in the bilingual digital library Bibliša, as one of the collections of aligned English-Serbian bi-texts [29,30], were also used in our approach.

A monolingual corpus from the mining domain was developed as part of a project related to managing mining project documentation using human language technology [31] and used within this research in the web and mobile applications.

### 2.3. General Purpose Morphological Dictionaries

Serbian has an extensive system of inflection and a complex agreement system that makes extraction of terminology more complicated, and thus the use of general purpose morphological dictionaries is indispensable for every lexicographic task [32].

An important lexical resource used for morphological analysis and extraction are the comprehensive electronic morphological dictionaries for Serbian (SrpMD) of simple- and multi-word units, covering general lexica, proper names, encyclopedic knowledge and terminology from a number of domains [33], with nearly 200.000 lexical entries. SrpMD entries include both a lemma and inflected forms supplied by grammatical information, semantic markers, domain information and relations of several types: derivational, lexical variation, component relations (between single words and terminological phrases).

For example, lexical entry *'rudar'* (miner, person engaged in mining, a worker in a mine) contains information related to part of speech: *'N'* (noun), morphological class *'N2'*, semantic tag *'+Hum'* (human), domain *'DOM = mining'*. Its inflected forms are: *'rudar'* (ms1v), *'rudara'* (mp2v:ms2v:ms4v:mw2v:mw4v), *'rudare'* (mp4v:ms5v), *'rudari'* (mp1v:mp5v), *'rudarima'* (mp3v:mp6v:mp7v), *'rudarom'* (ms6v), *'rudaru'* (ms3v:ms7v) where brackets show grammatical information: *'m'*—masculin, *'s'*—singular, *'p'*—plural, *'1–7'*—cases, *'v'*—animate.

The entry *'rudar'* is also related to the relational adjective *'rudarski'*, and appears as a component of several terminological phrases, for example, rudar na okresivanju (ripper), rudar na uglju (collier), rudar-podgrađivač (timberman), and so forth.

Over the past years, more entries related to raw material were added to SrpMD, which initially contained more than 3000 simple-word entries and 2000 multi-word entries from the raw material domain. The number of their morphological forms recorded in this resource is significantly larger. The simple-word forms pertaining to raw material terminology that have been processed and included in SrpMD [34] enabled further extraction of related terminological phrases according to the methodology described in [19]. Namely, for extraction to be effective, it is very important that the domain is relatively well covered with simple domain-specific words.

## 3. Resource Preparation

Preparation of resources is aimed at expanding and enriching available digital resources. These activities are not to be understood as one-time only activities, as each of them can be repeated periodically, when new opportunities for resource enrichment appear.

### 3.1. Dictionary (Retro)Digitisation

In order to expand and enrich the available digital resources, a number of paper dictionaries were digitised in the preparatory phase. After scanning, OCR and transformation to MS Word, with preservation of formats (bold, italic), manual correction was performed. The Word documents were then parsed, by a parsing procedure that was fine-tuned for each dictionary, according to its structure. Parsed data were finally transformed to structured formats: excel and xml, before being imported to the internal relational database. The procedure will be illustrated on one multilingual dictionary (MD) and one monolingual dictionary (DMMRT).

The digitisation and parsing of MD produced 16,491 term entries (examples of term entries are given in (Figure 1), where Serbian terms were aligned with one or more English term equivalents (the remaining 3 languages were also stored in the database, but they were not used in this approach).

**Figure 1.** Examples of scanned Mining dictionary entries.

The majority of dictionary entries (15,016) contained only one Serbian term, but there were 1355 entries with two terms, and 120 with 3–5 terms, resulting in a total of 18,092 Serbian terms, of which 16,916 distinct. As to the English part of the dictionary, there were 13,163 entries with one term, 2553 with two terms and 775 with 3–8 terms, resulting in a total of 20,878 English terms, of which 17,774 distinct.

Raw material terminology, akin to general technical terminology, contains a large number of multi-component terms. In the dataset obtained from the dictionary 23% of English entries are single word terms, 50% are two-component terms, 18% have three components and the remaining 9% have four or more. As for Serbian entries, 22% are one-component terms, 47% have two components, 17% have three, and the remaining 14% have four or more. The majority of English multi-compound terms are noun compounds. These linguistic constructions are most often composed of two or more nouns. for example, *'coal waste'—'jalovina'*, *'waste dump'—'odlagalište jalovine'*, *'gas pressure'—'pritisak gasa'*. However, they can also contain three, four or more nouns, for example, *'gas protection apparatus'—'lična zaštitna sredstva od gasova'*, *'mud circulation pressure hose'—'isplačno crevo'*.

Given the frequency of multi-component terms, an analysis of translational equivalents in English and Serbian was performed in terms of the number of their components. It was found that in 20% of cases both translational equivalents have one component, in 31% of cases both have two components, in 15% of cases the Serbian term has one component more than the English term, while in 13% of cases the English term has one component more, in 5% of cases the Serbian term has two components more, and in 3% of cases English has two components more. All other cases cover the remaining 13% of cases.

Entries in DMMRT have one or more senses per each term, described by a definition, and labeled by small letters *a*, *b*, *c*,. . . , *u*. Each individual sense can be related to one or more other terms in the dictionary, and it can be followed by its bibliographic source. Digitization of DMMRT yielded 28,757 terms with a total of 37188 sense definitions, where 24,115 terms have only one sense, 2942 have 2, 890 have 3, 641 have 4–6, 139 have 7–10, and 34 have 11–21. The most polysemous word is *'head'* with 21 senses, followed by *'drift'* and *'bottom'* with 20 senses. Types of relations between entries can be: See (4090), See also (3983), CF (compare, 1824), Ant (antonym, 20), Etymol. (etymology, 130), Syn: or syn.(synonym, 2532), Abbrev. (abbreviation, 77), etc. Figure 2) presents the entry *'accessory plate'* with five senses, marked by letters a-e. Two senses (a and e) are related to other dictionary terms (a to *'quartz wedge'* by CF, and e to three synonyms and two other terms by CF), and two senses (b and c) are followed by their source (Pryor).

**accessory plate**

a. The quartz wedge inserted in the microscope substage above the polarizer in order to estimate birefringence and to determine optical sign of uniaxial minerals. CF: quartz wedge b. The selenite plate that gives the sensitive tint of a specimen between crossed nicois. *Pryor, 3* c. The mica plate that retards yellow light. *Pryor, 3* d. In polarized-light microscopy, an optical device that may be inserted into the light train to alter light interference alter passage through, or reflection by, a crystalline material; e.g., quartz wedge, mica plate, gypsum plate, or Bertrand lens, e. In polarized-light microscopy, an optical compensator that may be inserted into the light train to alter birefringence after light passage through or reflection by an anisotropic material; e.g., quartz wedge, mica plate, gypsum plate, or Berek compensator. Syn: gips plate; glimmer plate; compensator. CF: Berek compensator; gypsum plate.

**Figure 2.** An example of scanned entry from DMMRT.

As to the components of the terms in DMMRT, 37% of the total terms are single word terms, 50% are two-component terms, 10% have three components and the remaining 3% have 4–7 components. Comparison with the English part of MD shows a similar pattern, as the percentage of two-component words is equal, while MD has 14% less one-component terms.

Additional 19 dictionaries from the raw material and related domains were digitized, parsed and stored in the database, adding 63,571 new entries. Five monolingual English dictionaries from the mining domain produced 5933 entries, three bilingual mining English-Serbian dictionaries produced 24,049 entries, three monolingual English dictionaries covering terminology from the mine safety domain contributed with 655 entries, and an English-Serbian dictionary of terminology in the field of waste management yielded 1968 entries. Dictionaries from related domains were also included, namely four English dictionaries producing 21,448 entries and three bilingual dictionaries producing 9518 entries.

One of the observations, even before this research started, was that several terms in paper dictionaries are not in use anymore. That observation initiated frequency calculation of Serbian terms in the mining corpus. Frequency in the corpus and the number of dictionaries that attest a term were the main criteria for post editing priority of the term.

Entries from all digitized dictionaries were stored in the same database, but in different structures, which correspond to their original data schema, and with reference to the original source. All of the structures can, in general, be mapped to the union of the structures of the two dictionaries presented in more detail, MD and DMMRT. Thus, a terminological entry in the common database can consist of a headword (list), rarely part-of-speech, equivalent(s) in other language(s), usually one, but sometimes more, labeled senses that include definitions, occasionally synonyms and abbreviations, links to other entries, bibliography, rarely specific domain.

### 3.2. Corpora Enlargement

The monolingual corpus of texts from the mining domain and related research work, which comprised 172 documents (in Serbian) with 2.7 million words in first release [31], was subsequently enlarged with 63 documents. The current version has 4.1 million words,

covering project documentation (26%), legislation (11%), doctoral dissertations (31%), textbooks and other mining literature (32%).

The bilingual corpus of texts aligned on the sentence level was produced from the bilingual digital library Bibliša. The initial set of 55 documents containing 4831 aligned Serbian-English sentences [29] was enlarged with 44 new documents containing 12,657 aligned sentences from the raw material and energy domains.

The crucial linguistic preprocessing steps within corpora enlargement are part-of-speech tagging and lemmatization. Part-of-speech tagging represents an automatic text annotation process in which words or tokens are marked by part of speech tags, which typically correspond to the main syntactic categories in a language (e.g., noun, verb). Lemmatization is the process by which inflected forms of a lexeme are grouped together under a base dictionary form. The Serbian corpus and the Serbian part of the bilingual corpus are tagged and lemmatized using a customised tagger [35], while the English part of the bilingual corpus is tagged by Treetagger [36,37].

Texts included in corpora are also processed using electronic dictionaries and local grammars. It is important to note that text processing and related mining vocabulary expansion is an iterative process. Namely, among other tasks, corpora are used for extraction of mining terminology, definitions and usage examples by applying different methods and tools.

### 3.3. Adding New Serbian Terms to General Lexica Dictionaries

Terminology from digitized dictionaries of raw material terminology in Serbian was checked by SrpMD and the corpus from the mining domain, for possible adding to SrpMD. We will illustrate this procedure by the results obtained from MD. The Serbian part of MD that contains headwords was transformed into a text, which was then analysed by SrpMD. Out of 12,655 different single words found in the text produced from the dictionary, 9758 were recognized by SrpMD. Among the 2897 (23%) that were not recognised, there were some acronyms (e.g., *'pH'*, *'RR'*, *'LD'*, *'TV'*), names (e.g., *'Western'*, *'Bets'*, *'Reni'*), archaisms (e.g., *'abanje'* instead of *'habanje'* (wear and tear), *'bolcn'* instead of *'zavrtanj'* (screw), etc.), as well as some OCR errors (despite manual check-up). Based on this analysis, a set of candidates for new entries into SrpMD were prepared (e.g., *'degazacija'* (degassing), *'eksploatabilan'* (exploitable), *'sabirnik'* (busbar), etc.). Each candidate was further checked against the mining corpus, and if the result (basically, its frequency) was satisfactory, it was added to the SrpMD.

The same procedure was applied to other dictionaries with Serbian entries. While the comprehensive terminological dictionaries (such as MD) contained a lot of simple words that were missing in SrpMD, smaller dictionaries, as expected, included frequently used terms that were mostly already in SrpMD. Thus, for example, in Electropedia 13% of words were not recognized by SrpMD, while in the Serbian part of the English-Serbian dictionary of terminology in the field of waste management 6% of words were not recognized. In all other dictionaries the percentage of unrecognized words was between 3%–5%, but whether they would be included into SrpMD depended on their frequency in the mining corpus.

Besides the digitized dictionaries, the Serbian corpus and the Serbian part of the bilingual corpus from the mining domain were yet another source of new raw material domain terms that did not exist in SrpMD. Extraction of simple words was relatively simple, namely, words that were not recognized by SrpMD were scrutinized, and if frequent enough, they became candidates for being added to SrpMD. Besides, less than 4% of words in the monolingual mining corpus were unrecognised by SrpMD, where approximately 1.3% out of these 4% were proper candidates to be added to SrpMD, the remaining unrecognized words being variables from equations (0.7%), acronyms (1%), low frequency (hapax and typos—0.5%), foreign names and words (0.5%).

However, when it comes to terms in the form of terminological phrases, their extraction from corpora becomes much more complicated. Automatic extraction of term candidates for Serbian relies on a procedure presented in [30,34]. Essentially, it is based on detecting

words in corpora that follow one of the 23 specific syntactic patterns, most frequent for noun terms (AN adjective-noun, NNg noun-noun in genitive case, AAN, . . . ). The first step in this task is to recognise and extract Serbian terminological phrases from the corpus using syntactic patterns, and calculate their frequency. Frequency was the main parameter for determining the rank of a terminological phrase as a candidate for processing for SrpMD. However, other measures of association, such as T-Score, Keyness, Log-likelihood, were also used, as described in detail in [30]. The task then proceeds by lemmatization of candidate terminological phrases, disambiguation for terminological phrases where more lemmas can be produced, and ends by production of the final lemma, which enables production of all inflected forms for each terminological phrase.

As in the case of single terms, frequency for terminological phrases was also calculated for each single-word component of the phrase, but for its lemma, not for the exact inflected form. Having in mind free word order in terminological phrases we were looking for a measure more loose than exact match. For each terminological phrase the following information is stored: minimum, average and maximum frequency of its components, number of "known" components-words recognized by SrpMD. Frequency in the corpus and the number of dictionaries that attest a term are the main criteria for post editing priority of the term.

For this paper, extraction of Serbian terminological phrases was performed with a frequency threshold of 10, and 12,632 candidate phrases were produced in lemmatized form. Frequency of each terminological phrase was calculated as the sum of frequencies of all its inflected forms. For example, *'kvalitet uglja'* (coal quality) has a frequency of 1110 as a sum of frequencies of its forms: *'kvalitet uglja'* (172), *'kvaliteta uglja'* (587), *'kvalitetom uglja'* (284), *'kvalitetu uglja'* (53), *'kvalitete uglja'* (8), *'kvaliteti uglja'* (2), *'kvalitetima uglja'* (4). Six most productive patterns, which produced 92% of candidates, are listed with examples and their frequencies:

- NNgi (32%), N2X—a noun followed by a word that does not inflect in the terminological phrase. Usually this word is a noun in the genitive or in the instrumental case; examples are *'kvalitet uglja'* (coal quality—1110), *'sistem upravljanja'* (management system—902), *'procena rizika'* (risk assessment—514).
- AN (29%), AXN—an adjective followed by a noun; the adjective and the noun have to agree in all four grammatical categories; examples are *'površinski kop'* (open pit—5738), *'ugljeni sloj'* (coal seam—1686), *'rudarski projekt'* (mining project—1412).
- NprepNp (11%), N4X—a noun followed by two words that do not inflect in the terminological phrase where these word form a prepositional phrase; examples are *'zdravlje na radu'* (occupational health—1323), *'čvrstoća na smicanje'* (shear strength—270), *'transporter sa trakom'* (belt transporter—240).
- N-N (10%), NXN—a noun followed by a noun that agrees with it in number and case, where the separator can be a hyphen; examples are *'gas-lift'* (197), *'blok dijagram'* (block diagram—192), *'bager vedričar'* (bucket excavator—174). This class had the largest number of recognized phrases for rejection, that is, those whose slightly different lemmas were already captured by another pattern, and this pattern should thus be placed with some lower priority in disambiguation.
- X-N (6%), 2XN—a noun preceded by a word that does not inflect in the terminological phrase. Usually it is a word that is used only in one or few terminological phrases, a prefix or an adverb derived from an adjective, while the separator can be a hyphen; examples are *'bto sistem'* (bto system—1728), *'pm preduzeće'* (pm company—373), *'y-osa'* (y-axis—19).
- NNgiNgi (4%), N4X—a noun followed by two words that do not inflect in the terminological phrase where these two words are adjectives/nouns in the genitive or instrumental case; examples are *'zaštita životne sredine'* (environment protection—668), *'eksploatacija mineralnih sirovina'* (mineral resource exploitation—228), *'efekat staklene bašte'* (greenhouse effect—109).

Evaluation follows, where the following is checked: is the extracted candidate a terminological phrase, which domain (mining, technical, etc.) and possibly subdomain it belongs to. If the domain or subdomain are identified, the appropriate semantic markers are assigned to the terminological phrase. After the evaluation process, all correctly evaluated terminological phrases were prepared for insertion into the terminological database Termi.

### 3.4. Adding Bilingual Terms

Bilingual lists of terms were considered a valuable resource in our approach, and they were generated from two sources, namely, by retrieval from the bilingual MD and by extraction from the aligned bilingual corpus.

Term entries from MD were parsed and only those that were confirmed by the mining corpus (monolingual or bilingual) were selected. As mentioned before, one term entry can comprise more terms (single or multi word) and confirmation for each term was looked for.

A total of 10,059 term entries from MD were retrieved, with sets of English terms aligned with sets of Serbian terms. The majority of them were subsequently marked by domain (24 different), subdomain (15) and semantic markers (35) as mentioned in Section 3.1. All markers used are subsets of markers—data category values in srpMD.

Bilingual terminology was extracted from the aligned bilingual domain corpus described in Section 3.2 using terminology extractors for Serbian and English, and Bilte [38]), a tool for chunk alignment [39,40]. The method combines the approach with existing domain terminology lexicons with term extraction tools. For English, FlexiTerm [41] was used with threshold 3 and TermSuite [42] with threshold 4, based on the experience from other domains and the fact that they use different linguistic filtering. A total of 8456 term candidates for English were selected. For Serbian, the same shallow parser was used as in the case of monolingual extraction (Section 3.3), as well as the same calculation of termhood, a frequency-based measure, which qualified 7825 candidates as terms.

Monolingual lists of extracted terms were further expanded by terms retrieved from digitized dictionaries yielding 94,539 English terms and 48,096 Serbian terms. Some terms were found in both datasets: extracted from text and retrieved from dictionaries, namely, a total of 2285 English and 308 Serbian terms.

The GIZA++ [43] and Moses toolkit [44] for statistical machine translation (SMT) were used for word alignment. Aligned chunks, presented in the so-called phrase table, are obtained as output from Moses, together with their phrase translation scores. After pruning the phrase table with the threshold probability of 0.85, the remaining chunks were lemmatized and further filtered to select those in which both parts of the pair contain a candidate term from the raw material domain. More details about options and the procedure are available in [40]. The output of this phase contained 8202 Serbian-English pairs as term candidates whose English part was confirmed and 3605 where both language parts were confirmed. In the first step, candidates that were found in digitized dictionaries, or were already assessed as terms, were automatically confirmed, but candidate pairs had to be inspected manually, which yielded a list of 2737 term pairs. General terms, such as, *'red' (row)*, *'kompozicija' (composition)*, *'din' (dinar)*, *'minimalan' (minimum)*, *'izvor informacija' (source of the information)*, . . . were excluded, as well as those wrongly aligned, such as: *'naftovod' (pipeline oil)*, *'mreža' (telephone network)*, *'deponija' (deposit)*, *'oblik poklopca' (shape of the cover)*, . . . A wider set of terms will be evaluated in the near future.

For evaluation of bilingual candidates, besides frequencies for single terms, we have also used a heuristic for evaluating terminological phrases based on the following observations. The last noun in English noun compounds, which represent the majority of English terminological phrases, as a rule, is the head word carrying the basic meaning, while the preceding nouns are narrowing this meaning, that is, behaving like adjectives. The meaning of a noun compound in English thus flows from right to left, but the Serbian translational equivalent cannot be formed analogously, namely, by a sequence of corresponding Serbian nouns. Thus, within the analysis, the most frequent constructions used as Serbian translational equivalents for English noun+noun compound were determined:

- noun + noun in the genitive (e.g., *'coal mining'*-*'eksploatacija uglja'*)
- adjective + noun (*'waste water'*-*'otpadna voda'*)
- noun + prepositional phrase (*'belt conveyor'*-*'transporter sa trakom'*)
- paraphrase (*'crusher stower'*-*'mašina za drobljenje i pneumatsko zasipanje'*)
- one-word name (*'crushing machine'*-*'drobilica'*).

This heuristic was used to select the most promising candidates among the extracted bilingual terminological phrases.

As in the case of multilingual terms and terminological phrases, after the evaluation process, all correctly evaluated bilingual terms were prepared for insertion into the terminological database Termi. So far, more than 3000 term-to-term pairs were inserted. In this process they were merged to form synonymous sets (synsets) by using information from existing dictionaries and simple rules, such as: if two English terms are translated by the same Serbian term they are candidates for synonyms.

## 4. Terminology Aggregation and Presentation

### 4.1. Data Integration Procedure—The Pipeline

The main goal of our approach is to merge and link all available terms in the raw material domain into one lexicon structure, within the terminological database Termi and as linguistic linked data available via SPARQL endpoint, in the first place by aligning as much as possible term entries from dictionaries and other resources covering raw material domain terminology. Besides the aim of aggregating terms from different resources, one of the reasons for alignment of terms from multiple dictionaries (paper and electronic) was to assess term usage, which determines its importance for raw material terminology. On the other hand, alignment of terms with SrpMD was necessary, since these dictionaries are a base resource for lemmatization and multiword term extraction. Since SrpMD are already in the lexical database Leximirka [32], developed and managed by the same research team, this type of alignment was possible.

Figure 3 presents an outline of the pipeline for termbase population, which starts with collecting and preparing research papers, project documentation, and textbooks in Serbian for the monolingual corpus and aligning English-Serbian texts for the bilingual parallel corpus. Also, paper dictionaries, both monolingual and bilingual are digitized, parsed and stored in an auxiliary database as structured data in XML format.

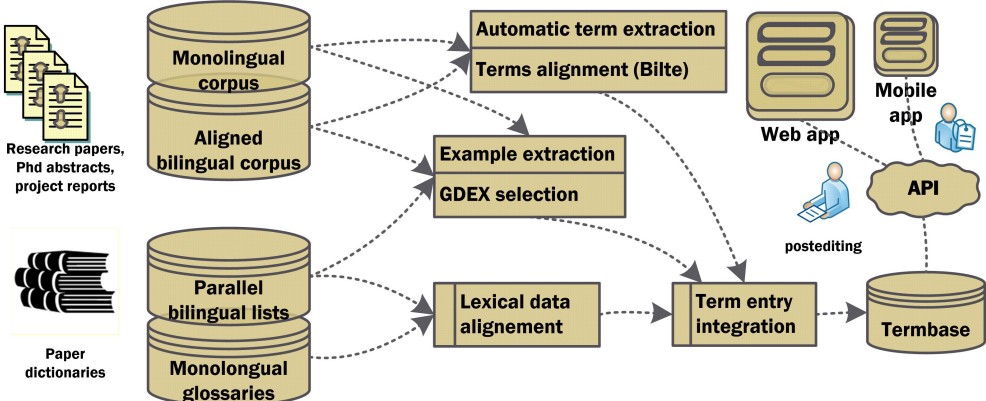

**Figure 3.** The pipeline for terminology compilation (termbase population).

Compiled resources also comprise monolingual lists derived from all available resources, interlinked with their source entries, for example Serbian list from Serbian monolingual dictionaries and Serbian part of bilingual dictionaries. Translation equivalents are retrieved from bilingual dictionaries and within the word alignment phase (more in Section 4.2), keeping again information about the original dictionary source.

Extracted terms were also subject to a labeling procedure, which we will illustrate here on the example of MD. Out of 16,491 entries obtained from MD, 12,018 (73%) were

manually classified and markers for domain and subdomain, as well as semantic labels, were assigned to them. The remaining 4473 (27%) unclassified entries included words from general lexica and some rarely used terms. The classified entries are mostly from the mining domain, more precisely, there are 4793 (40%) entries common for different areas of mining. The basic vocabulary from related domains is also included, for example, 2398 (20%) entries related to geology, hydrogeology and geography, 860 (7%) entries related to transport, rock mechanics, surveying, environment protection, safety, construction, transport and electrical engineering, while 3082 (26%) entries belong to the general technical terminology. There are also entries from basic science, for example, 885 (7%) terms related to biology, chemistry, mathematics, informatics and physics.

Among entries from the mining domain, those related to a specific subdiscipline of mining were identified by mining experts, and marked by a subdomain marker, as for example, entries related to mineral processing (251), transport (243), or underground mining (469). Additional semantic labels were also assigned, for example, material (699), device (536), machine (384), mineral (313), facility (288), instrument (279), etc.

The part-of-speech was semi-automatically assigned, where only 40 entries were marked as adjectives, 250 as verbs, and all other as nouns.

Lexical entry alignment with DMMRT is performed using terms on the English side of the MD. Since one English term can have several senses, such alignments are marked for manual filtering. An indicator is used for status: automatic relation or manually evaluated.

A terminological dictionary must accompany each entry with a scientifically and lexicographically correct definition [45]. There are very few such dictionaries in the Serbian language, as most of the published Serbian terminological dictionaries are only translational (bilingual or multilingual). An ongoing activity is the adaptation of English definitions, which are the most comprehensive in DMMRT, to Serbian, in the post-editing phase, where priority is given to the most frequent terms, both in the corpora and in the dictionaries.

Finally, candidates are harmonised and assembled to the microstructure of the lexical database Termi, which consists of a headword, synonyms, abbreviations, definition, for each language, bibliographic source and possibility to include illustration and other external content. Term entries in Termi are organised into a hierarchical structure, and additional relations between entries are envisaged, but still not implemented. Automatic hierarchical positioning was based on subdomain and semantic markers, but it is subject to repositioning in the post-editing phase.

Information integration beyond the level of individual dictionaries and across the language resource community has become an important concern, and the most promising technology to achieve this goal is to adopt the Linked (Open) Data (LOD) paradigm for publishing lexical resources, that is, to use URIs for unambiguously identifying lexical entries, their components and their relations in the web of data—to make lexical datasets accessible via http(s), to publish them in accordance with W3C-standards such as RDF and SPARQL, and to provide links between lexical data sets and with other LOD resources [46].

In our research we were also aiming at compatibility with the Linked Data approach, using its set of design principles for sharing machine-readable interlinked data on the Web. This vision of globally accessible and linked data on the internet is based on RDF standards of the semantic web, using RDF serialisation for data representation. To that end, our approach envisages export of lexical database data in RDF that is compliant with the *The OntoLex Lemon Lexicography Module* [47], lexicog [48], as an extension of Lexicon Model for Ontologies (lemon) [49,50]. This is also in line with activities within NexusLinguarum COST action [51], which promotes synergies across Europe between linguists, computer scientists, terminologists, language professionals, and other stakeholders in industry and society, in order to investigate and extend the area of linguistic data science. An example of RDF export is presented in Figure 4 followed by the Turtle RDF Syntax [52] to illustrate the use of the model.

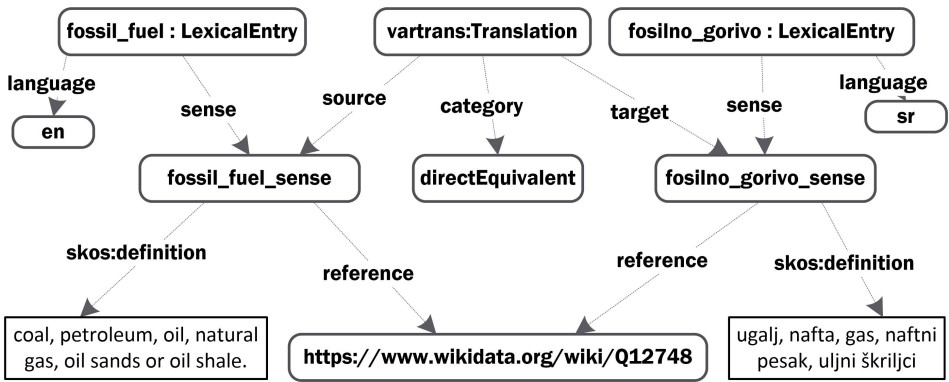

**Figure 4.** The graph for the translation of lexical entries: *'fossil fuel'-'fosilno gorivo'*).

```
:fossil_fuel a ontolex:LexicalEntry;
   dct:language <http://lexvo.org/id/iso639-1/en> ;
   lexinfo:partOfSpeech lexinfo:noun;
   ontolex:lexicalForm :fossil_fuel-form;
   ontolex:sense :fossil_fuel_sense.
:fossil_fuel-form a ontolex:Form;
   ontolex:writtenRep "fossil fuel"@en.
:fossil_fuel_sense  skos:definition "coal, oil, gas, oil sands or oil shale"@en;
   ontolex:reference <https://dbpedia.org/page/Fossil_fuel>;
   ontolex:reference <https://www.wikidata.org/wiki/Q12748>;
   ontolex:reference <http://eurovoc.europa.eu/6045>.

:fosilno_gorivo a ontolex:LexicalEntry;
   dct:language <http://id.loc.gov/vocabulary/iso639-1/sr> ;
   lexinfo:partOfSpeech lexinfo:noun;
   ontolex:lexicalForm :fosilno_gorivo-form;
   ontolex:sense :fosilno_gorivo_sense.
:fosilno_gorivo-form a ontolex:Form;
   ontolex:writtenRep "fosilno gorivo"@sr.
:fosilno_gorivo_sense skos:definition "ugalj, nafta, gas, naftni pesak ili
   uljni škriljci"@sr;
   ontolex:reference <https://www.wikidata.org/wiki/Q12748>.

:trans_fossil_fuel_sense-fosilno_gorivo_sense a vartrans:Translation;
      vartrans:source :fossil_fuel_sense;
      vartrans:target :fosilno_gorivo_sense;
      vartrans:category
         <http://purl.org/net/translation-categories#directEquivalent>.
```

Further details related to the above example, namely, the novel module for frequency, attestation and corpus information (FrAC) [53] is described in the next section.

### 4.2. Dictionary Examples and Frequencies

None of the dictionaries we have used contain examples of term usage. Our intention was to select actual terms that can be found in domain texts and to link usage samples to both monolingual and bilingual terms entries. Previous (and actual) practice in Serbian lexicography has relied on retrieving example candidates and definitions manually from different online sources and printed material (over a number of years), but it is evident that a more systematic and corpus-evidence-based approach was needed.

A method for the selection of good examples for Serbian terms was developed based on a feature extraction web services and knowledge retrieved from SASA Dictionary as the Gold Standard for Good Dictionary Examples (GDEX) for Serbian [54]. The method is based on a detailed analysis of various lexical and syntactic characteristics of examples in published dictionaries. The initial set of functions was inspired by a similar approach

for other languages. The distribution of the characteristics of examples from this corpus is compared with the characteristics of the distribution of the sample sentences extracted from the corpus that contains different texts. The approach was adapted to work also for English and to be applied for bilingual aligned sentences. For ranking, we have used a weighted score derived from lexical features (e.g., sentence length, number of all no space chars, digits, weird chars, commas, full stops, punctuation, number of all tokens, average token length, max token length, sentences between 15 and 40 tokens, ...), word-based features (e.g., number of words, capitalised words, ...) and other features (e.g., average frequency in corpus, number of stop words, proper names, pronouns). New features were introduced for bilingual examples, for example, difference in sentence length measured in words, where examples in which a sentence in one language is short and in the other language long are avoided. An example containing terms as key words in context in English and Serbian, sentence examples and calculated features is:

```
109867|7.2011.60.8|7.2011.60.8_n44|Fossil fuel|Fosilno gorivo|Carbon emissions
from sources other than fossil fuel combustion are now incorporated in the
National Footprint Accounts.|Emisije ugljenika iz drugih izvora, ne samo iz
sagorevanja fosilnih goriva sada su ubeležene u Izveštaje o nacionalnoj stopi
emisije zagadenja.|120|104|0|37|0|1|1|True|18|5.778|12|True|True|True|True|17|
6.0588|12|2|3|0.0|7|145|124|0|52|1|1|3|False|23|5.392|12|True|True|True|True|
20|5.5|11|1|1|10955.428|7
```

For entries with no examples in the bilingual corpus, monolingual examples were extracted from the Serbian mining corpus. Apart from offering preselected examples, it is important to enable the user to browse the concordances for a lemma, as well as syntactic patterns, as presented in the next section in Figure 5.

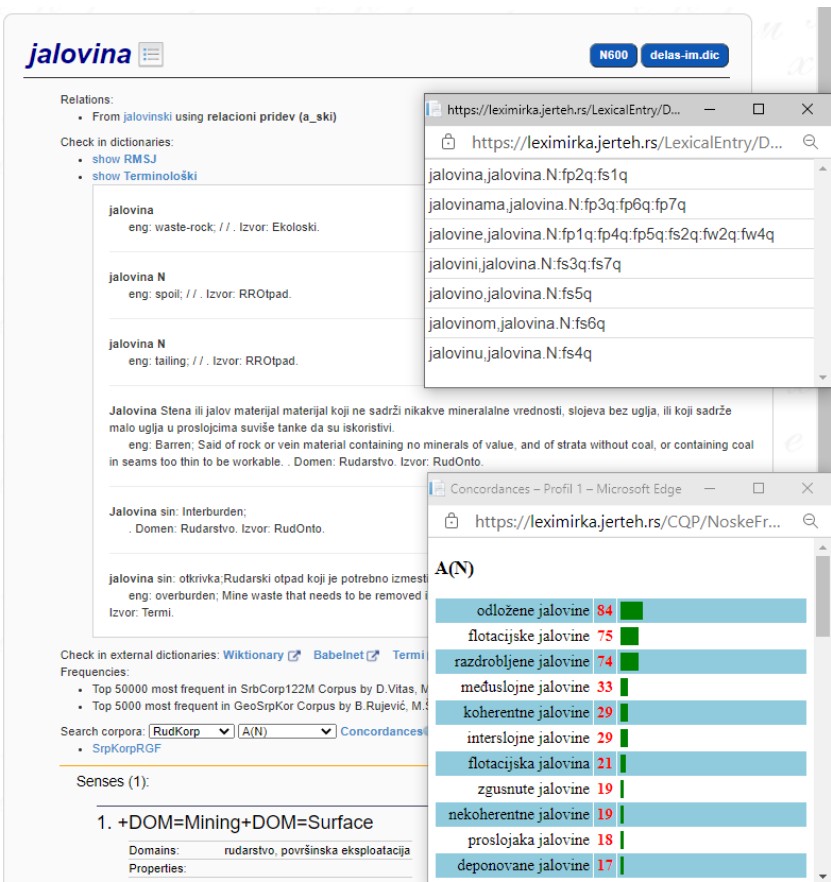

**Figure 5.** The Leximirka app for lexical database management.

Relative frequency (normalized per million) is assigned to terms from the mining corpus (as domain specific) and for the corpus of standard Serbian (as reference), in order to calculate the so-called keyness score, which is expected to represent the extent of the frequency difference.

Frequency information is a crucial component in human language technology, so the FrAC module includes terminology to capture such information, in order to facilitate sharing and utilising this valued information [53]. Sketch engine API [55,56] is used for calculation of frequencies, for word-sketch retrieval with collocations and for thesaurus with related words association measures (Statistics used in the Sketch Engine [57,58]). The Python script prepared in the form of a jupyter notebook was published at github [57]. Current work of the Ontolex group is focused on modeling word embeddings, collocations and similar words and we will add this feature when it becomes stable. An example of ontolex-lemon frequency and attestation snippet is:

```
# subproperty definition for frequency in mining corpus
:rudkorFrequency rdfs:subClassOf frac:CorpusFrequency .
:rudkorFrequency rdfs:subClassOf [
   a owl:Restriction ;
   owl:onProperty frac:corpus ;
   owl:hasValue <https://app.sketchengine.eu/#
      dashboard?corpname=user%2FAleksandraTomasevic%2Frudkor>] .
# frequency assessment (in mining corpus)
:fosilno_gorivo frac:frequency [
   a :rudkorFrequency;
   rdf:value "38"^^xsd:int].

# usage examples as attestations
:fosilno_gorivo frac:attestation attestation_1324567;
attestation_1324567 a frac:Attestation ;
   cito:hasCitedEntity   <https://app.sketchengine.eu/#
      dashboard?corpname=user%2FAleksandraTomasevic%2Frudkor> ;
   rdfs:comment "Dokument 31, DK_Monitoring u zivotnoj sredini" ;
   frac:locus :locus_2415677;
  frac:quotation "Koncentracija zagađujućih supstanci, posebno
   onih koje se izdvajaju sagorevanjem fosilnih goriva, varira
   u odnosu na godišnje doba (leto, zima)." .
:locus_2415677 a :Occurrence ;
   nif:beginIndex 80 ;
   nif:endIndex 96.
```

We have just started using VocBench, a web-based, multilingual, collaborative development platform for managing Ontolex-lemon lexicons among other RDF datasets [59], for publishing terminology as RDF data, in order to meet the needs of semantic web and linked data environments. VocBench is an open source web platform for collaborative development of datasets in compliance with Semantic Web standards, offering a general-purpose collaborative environment for development of any type of RDF dataset (with dedicated facilities for ontologies, thesauri and lexicons), including editing capabilities and managing SPARQL endpoint [60]. The system is able to interact with standard technologies in the RDF/Linked Data world, with the possibility to surf linked open data on the Web, access SPARQL endpoints, resolve RDF descriptions through HTTP URIs, and so forth, as well to import/export data through standard Graph Store APIs and the like.

### 4.3. The Web and Mobile App

The application for management of Serbian morphological dictionaries, including the evaluation of automatically extracted term candidates used in this approach is Leximirka [61]. Figure 5 presents a web page with term entry *'jalovina'*, where the user can see (1) inflected

forms with grammatical categories, (2) inflectional class (*'N600'*) and dictionary (*'delas-im.dic'*); (3) dictionary entries from other dictionaries (digitized and digitally born) grouped by dictionary type (descriptive, terminological, bilingual); (4) related entries (e.g., relational adjectives *'jalovinski'*), lexical variants, derived terms; (5) corpus frequencies; (6) corpus selection with links to concordances and frequency histograms for simple lemma query or predefined syntactic patterns (in figure pattern AN where N is the headword *'jalovina'*), (7) one or more senses with semantic and domain markers.

An important feature of this system is the possibility to insert a formula in the definition, which is often necessary to precisely define a concept. The Figure 6 presents a part of the screen with a latex form of definition and its preview on the same panel. The JavaScript display engine for mathematics MathJax [62,63] that works in all browsers is used in the web application, and KaTeX [64,65] for formula rendering in the mobile application.

Parametar koji karakteriše otpornost na usitnjavanje određene čvrste materije. Određuje se pomoću laboratorijskog mlina sa šipkama ili kuglama po Bondovoj metodi. U industrijskim uslovima on se određuje po jednačini: $W_{i}=W_{t}\times \frac{\sqrt{F}}{\sqrt{F}-\sqrt{P}}\times \sqrt{\frac{P}{100}}$. Koristi se za proračun snage pogonskih motora i izbor drobilica i mlinova za usitnjavanje.

Parametar koji karakteriše otpornost na usitnjavanje određene čvrste materije. Određuje se pomoću laboratorijskog mlina sa šipkama ili kuglama po Bondovoj metodi. U industrijskim uslovima on se određuje po jednačini:

$$W_i = W_t \times \frac{\sqrt{F}}{\sqrt{F}-\sqrt{P}} \times \sqrt{\frac{P}{100}}$$. Koristi se za proračun snage pogonskih motora i izbor drobilica i mlinova za usitnjavanje.

**Figure 6.** Formula editing and preview in term entry.

The mobile application allows the user to search for a Serbian or English term, where the query is submitted to the Termi API and a list of entries is retrieved, with a further possibility to request examples for selected entries. Figure 7 presents screenshots of mobile and web applications.

Besides for search, browse and the described export, the application can also be used for preparation of a dataset for Lexonomy [66,67]. Figure 8 presents a panel for term entry editing, which is connected with the Sketch-engine and enables retrieval of examples from a related corpus, in our case the corpus from the mining domain.

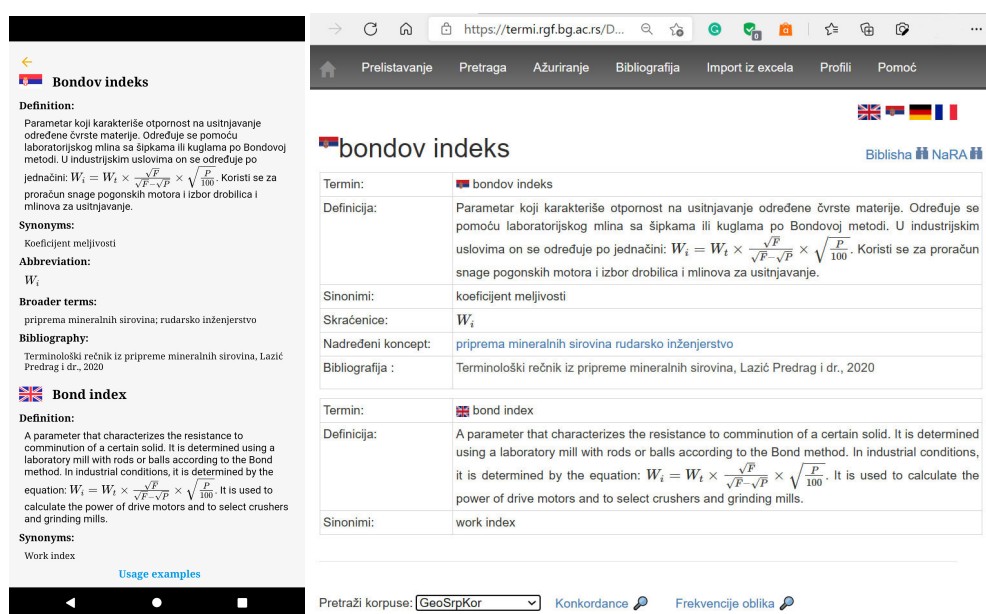

**Figure 7.** The mobile and Termi web application data entry preview for term entry.

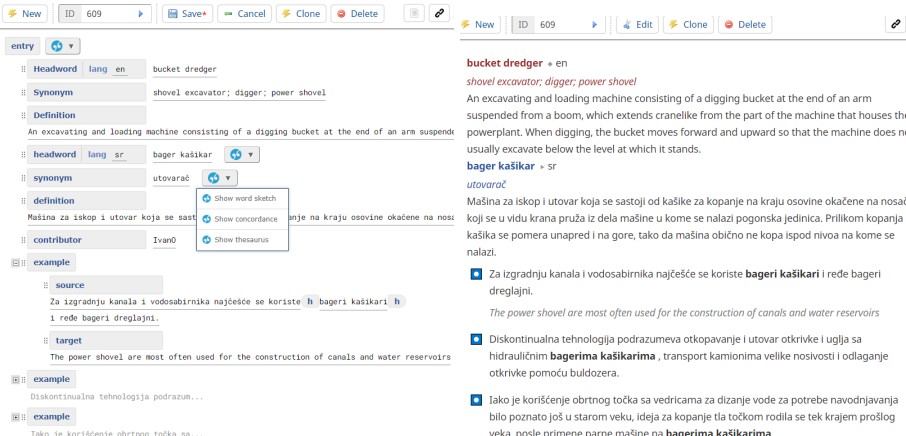

**Figure 8.** The Lexonomy data entry editing and preview for term entry.

## 5. Discussion

The presented approach to the development of terminology for the raw material domain, based on digitized and electronic dictionaries, terminological and domain corpora enables systematic development of terminology, complementing traditional terminological dictionaries with usage examples, and providing a comprehensive picture of the use of terms in various dictionaries, textbooks, professional and scientific literature. A terminology system that includes a relational terminology database, a SPARQL endpoint with linguistic linked open data, on the one hand, and a web and mobile application, on the other, provides a technological solution that enables data management, continuous updating, upgrading and expansion of available data, while various application forms (web and mobile) make the content more accessible to users.

Integration of terminology with the lexical database and morphological dictionaries, which enables support for a complex inflectional system, is important for all languages with rich morphology, such as Serbian. Integration with corpora, both standard and terminological, provides insight into the use of terms in modern language and in a specialized domain, enabling insight into individual examples, but also into the frequency of use of different syntactic structures, enabling research into collocations of individual terms.

The approach is demonstrated on the example of mining, but the same approach and developed software solutions can be used for other areas, which is certainly one of the further directions of activity. It should also be noted that the approach can be applied to other languages, depending on the available data and not on the language itself.

The vast amount of digitized resources, 22 dictionaries, monolingual corpus with 4 million words and bilingual with 12,657 aligned sentences, represent the basis for numerous other research activities, development of collocation dictionaries, creation of possibly printed dictionaries of different volumes (including pocket and encyclopedic ones). Such a system will make it easier for students to translate from English with the use of correct terms in Serbian, but also when writing articles and translating into English for academic purposes.

Since the presented approach used a combination of reuse of data, automatic extraction and manual post-editing, a comparison of those aspects with some similar solutions follows.

When it comes to the reuse of data, we followed the idea of the Sõnaveeb language portal of the Institute of the Estonian Language [10], which contains data from 70 dictionaries and termbases, comprising a total of 200,000 Estonian headwords with many new types of lexicographic information: collocations, etymology, multi-word expressions, and so forth. The number of lexicons in our case is much smaller, but at the moment we are focused on the mining domain and related terminology. Also, our system does not include etymology, but we plan to introduce it in the future. There is a difference in the software solution for mobile users, as Institute of the Estonian Language decided to produce a responsive web page that adapts to different devices by automatically adapting to the screen, whether it is

a desktop, laptop, tablet or smartphone, while we produce a mobile android application akin to Oxford Dictionary or Merriam-Webster. Finally, the difference related to corpus use is that our system has direct connection with corpora, both domain and general language, which allows users to retrieve concordances, collocations defined by syntactic patterns and graphical frequency presentations. The Sõnaveeb project is a result of several projects in a longer period, developed by a much bigger team, but we are following their ideas to continually improve our system.

An Integrated Approach to Biomedical Term Identification Systems [11] combines several sources of information and knowledge bases to provide biomedical term identification systems with modular architecture, which includes medical term identification, retrieval of literature and ontology browsing by applying several NLP technologies. The similarity with our system is in combining several terminological and lexical resources, as well as the use of various NLP techniques, while the difference is that their system generates a conceptual graph that semantically relates all the terms found in the text, which would be our plan for future research. On the other side, our system is building a new resource that integrates a number of digitized and electronic resources.

The corpus-based approach for extracting domain-oriented and technical words applied to improve the efficiency of corpus analysis in COVID-19 big textual data [7] is based on elimination of function words and meaningless words. This, widely accepted, approach for information retrieval is not so successful for knowledge extraction, lexicographic and terminological purposes, so we are relying on a combination of syntactic patterns [34,42,68] and statistical association measures for domain terms: log-likelihood [69], c-value/nc-value [70], because such hybrid systems have proved to yield the best solutions [71].

Besides monolingual term extraction, we also followed a different approach when it comes to bilingual term extraction [72,73]. We first perform monolingual extraction of domain-specific terms, using available terminology extractors, and then, given a source term and a parallel sentence pair in which it appears, a set of possible translations are obtained. There are different options: to use automatic translation, trained on the same corpus using GIZA++ [40,43], to apply a word aligner [72], or to use log-likelihood comparison and phrase-based statistical machine translation models as in TermFinder [73]. We rely on previous research [27,39,40] that proved successful for bilingual term extraction in other domains, where one language is Serbian.

The Sketch-engine [9] has different types of extraction implemented, for various languages, starting with keyword extraction, word sketches, usage examples, and thesaurus, but it is not fully adapted for Serbian, and its results are far less successful than those obtained in our research [40,68]. Sketch Engine offers tools to significantly speed up the process of dictionary building, especially the "OneClick Dictionary" process, which consists of generating a headword list, providing part-of-speech labels, usage labels, generating candidates for example sentences, collocations, synonyms and thesaurus entries, definitions and/or translations [74]. The output is pushed into the Lexonomy dictionary writing system [66,67], from where lexicographers can communicate with the Sketch Engine during the post-editing phase, enabling browsing of concordances from a corpus and retrieval of selected examples directly into the interface form. The integration with corpus is a rare and very useful possibility, but Lexonomy lacks hierarchy browsing, mathematical formulae are not supported and search capabilities are limited.

## 6. Conclusions

The presented approach relies on the results of previous research in the field of NLP and terminology, but represents the first comprehensive solution for both building and using a terminology system that includes data, application and user interface layers covering different data and software technologies.

The automation of data publishing in the form of linked data, as one of the core pillars of the Semantic Web or the Web of Data, provides links between data sets that are

understandable not only to humans, but also to machines, by sharing machine-readable interlinked data on the Web.

The next big challenge for the future is the automation of core lexicographic tasks related to semantics, such as finding definitions or identifying senses in two distinct processes: word-sense disambiguation (attributing the correct sense from a predefined set of senses) and word-sense induction (clustering of senses based on word context). Also, integration of results into linked open data especially word embeddings, collocation and similarities.

In future research we will incorporate synonyms for lexical sememe (smallest semantic unit for describing real-world concepts) prediction using an attention-based model [75], which scores candidate sememes from synonyms, by combining distances of words in the embedding vector space, and derives an attention-based strategy to dynamically balance two kinds of knowledge from a synonymous word set and word embedding vector.

**Author Contributions:** Conceptualization, O.K. and R.S.; Data curation, A.T. and L.K.; Formal analysis, R.S. and M.Š.; Investigation, O.K.; Methodology, O.K., R.S. and I.B., Validation, A.T. and L.K.; Software, O.K., M.Š. and I.B.; Writing—all authors. All authors have read and agreed to the published version of the manuscript.

**Funding:** This research was funded by Finnish Work Environment Fund and Ministry of Education, Science and Technological Development Republic of Serbia within European Science Program SAFERA (European Research on Industrial Safety towards Smart and Sustainable Growth) grant SafePotential, for period 2019–2020. Access to SketchEngine and Lexonomy is provided by the ELEXIS project funded by the European Union's Horizon 2020 research and innovation programme under grant number 731015. Linked data development is supported by the COST Action CA18209-NexusLinguarum "European network for Web-centred linguistic data science".

**Institutional Review Board Statement:** Not applicable.

**Informed Consent Statement:** Not applicable.

**Data Availability Statement:** The data presented in this study are freely available for the search on site: https://termi.rgf.bg.ac.rs/ (accessed on 15 March 2021) and on request from the corresponding author.

**Acknowledgments:** The authors thank Ivan Obradović for proofreading and constructive comments, Cvetana Krstev for use of electronic dictionary of Serbian, Petar Popović for corpus management and Branislava Šandrih for feature extraction from usage examples.

**Conflicts of Interest:** The authors declare no conflict of interest. The funders had no role in the design of the study; in the collection, analyses, or interpretation of data; in the writing of the manuscript, or in the decision to publish the results.

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
