# Peer review of "A Data Driven Approach for Raw Material Terminology"

_applsci, doi:10.3390/app11072892_

Round 1

Reviewer 1 Report

The presented work concerns the current problem of adapting resources to the needs of modern systems supporting autonomous agent systems.

Creating knowledge bases that are multilingual, interoperable, reusable and semantically machine-understandable is, on the one hand, an almost historical necessity, and, on the other hand, a practical challenge.  

The authors describe their experiences with just such a practical solution, which is interesting for the reader, and from the scientific point of view: it is a very important contribution to the development of ontology and formalization of knowledge in the aspect of machine processing.

As the Authors said: solution for both building and using a terminology system that includes data, application and user interface layers covering different data and software technologies.

The presented manuscript is interesting and well prepared. 

Author Response

Reviewer 2 suggested to add the comparison of the proposed solution with other similar solutions in the field. In our approach we have used a combination of reuse of data, automatic extraction and manual post-editing, so we tried to cover comparison on all those aspects. Section 5 “Discussion and Conclusion” was split into section 5 “Discussion” and section 6 “Conclusion” and the recommended comparison was inserted in Section 5 “Discussion”, in lines 769 - 826, coloured in blue.

The second comment of the Reviewer 2 was related to spelling mistakes, so we double checked the text and made minor changes.

Reviewer 2 Report

This paper presents an interesting approach that draws on the use of opportunities offered by electronic lexicography, as well as various available natural language processing (NLP) techniques, to develop a semi-automatic system for dictionary production.

The proposed approach can be applied to other languages, depending on the available data and not the language itself, which can be a great advantage.

I would recommend a comparison of the proposed solution with other similar solutions in the field.

The paper has some spelling mistakes.

Author Response

We thank the Reviewer 2 for the suggestion to add the comparison of the proposed solution with other similar solutions in the field.

In our approach we have used a combination of reuse of data, automatic extraction and manual post-editing, so we tried to cover comparison on all those aspects. Section 5 “Discussion and Conclusion” was split into section 5 “Discussion” and section 6 “Conclusion” and the recommended comparison was inserted in Section 5 “Discussion”, in lines 769 - 826, coloured in blue.

Qualitative comparison is given on several aspects, but quantitative comparison for the whole process is not appropriate. For bilingual term extraction we are referring to previously published with research with, precision results.

The second comment of the Reviewer 2 was related to spelling mistakes, so we double checked the text and made the following changes:

“wordlist” into “word list”,

“encyclopaedias” with more frequent variant “encyclopedias”,

“digitalisation” to “digitisation” since the digitization refers to the conversion of specific products from the analogue to the digital format, while digitalization is a generic term for the use of digital technologies and of data in order to create revenue, improve business, replace/transform business processes.

“general dictionaries served as control dictionary languages” changed to “general language dictionaries served as control dictionaries”.

“fullstops” changed to “full stops”

“nospace” changed to “no space”

“stopwords” changed to “stop words”

“noun compounds (e.g. nickname)” changed to “compound nouns (e.g. nickname)”

“complex verbs” changed to “phrasal verbs”

“ESP (English for specific purposes)” changed to “ESP (English for Specific Purposes)”

“Dictionary entries may be linked to original texts in corpora as examples of usage.” changed to “Examples of usage may be extracted from original texts and linked to dictionary entries.”

“options of search” changed to “search options”

“ and and ” changed to “ and ”

“ecology and other disciplines, being.” Changed to “ecology and other disciplines.”

“digitalized” changed to “digitized”

“hand crafted” changed to “handcrafted”

“in mining engineering in specific sub-fields” changed to “in specific subfields of mining engineering”

“relations of several types: derivations, lexical variation” changed to “relations of several types: derivational, lexical variation”

“digitised” changed to “digitized”

“prepositional expression” changed to “prepositional phrase”

“which consist of” changed to “which consists of”

“For entries where there were no examples” changed to “For entries with no examples”

“ i order” changed to “ in order”

- few determiners “the” were missing
